# Bioactive Carbonate Apatite Cement with Enhanced Compressive Strength via Incorporation of Silica Calcium Phosphate Composites and Calcium Hydroxide

**DOI:** 10.3390/ma16052071

**Published:** 2023-03-03

**Authors:** Arief Cahyanto, Michella Liemidia, Elin Karlina, Myrna Nurlatifah Zakaria, Khairul Anuar Shariff, Cortino Sukotjo, Ahmed El-Ghannam

**Affiliations:** 1Department of Dental Materials Science and Technology, Faculty of Dentistry, Padjadjaran University, Jl. Raya Bandung Sumedang KM 21, Jatinangor 45363, Indonesia; 2Department of Restorative Dentistry, Faculty of Dentistry, University of Malaya, Kuala Lumpur 50603, Malaysia; 3Biomaterials Technology Research Groups, Faculty of Dentistry, University of Malaya, Kuala Lumpur 50603, Malaysia; 4Faculty of Dentistry, Padjajaran University, Jl. Raya Bandung Sumedang KM 21, Jatinangor 45363, Indonesia; 5Department of Endodontology and Operative Dentistry, Faculty of Dentistry, Universitas Jenderal Achmad Yani, Jl. Terusan Jenderal Sudirman, Cimahi 40531, Indonesia; 6School of Materials and Mineral Resources Engineering, Engineering Campus, Universiti Sains Malaysia, Nibong Tebal, Pulau Pinang 14300, Malaysia; 7Department of Restorative Dentistry, University of Illinois at Chicago, Chicago, IL 60612, USA; 8Department of Mechanical Engineering and Engineering Science, University of North Carolina at Charlotte, Charlotte, NC 28223, USA

**Keywords:** compressive strength, bioactivity, carbonate apatite, silica calcium phosphate composite, calcium hydroxide, SBF

## Abstract

Carbonate apatite (CO_3_Ap) is a bioceramic material with excellent properties for bone and dentin regeneration. To enhance its mechanical strength and bioactivity, silica calcium phosphate composites (Si-CaP) and calcium hydroxide (Ca(OH)_2_) were added to CO_3_Ap cement. The aim of this study was to investigate the effect of Si-CaP and Ca(OH)_2_ on the mechanical properties in terms of the compressive strength and biological characteristics of CO_3_Ap cement, specifically the formation of an apatite layer and the exchange of Ca, P, and Si elements. Five groups were prepared by mixing CO_3_Ap powder consisting of dicalcium phosphate anhydrous and vaterite powder added by varying ratios of Si-CaP and Ca(OH)_2_ and 0.2 mol/L Na_2_HPO_4_ as a liquid. All groups underwent compressive strength testing, and the group with the highest strength was evaluated for bioactivity by soaking it in simulated body fluid (SBF) for one, seven, 14, and 21 days. The group that added 3% Si-CaP and 7% Ca(OH)_2_ had the highest compressive strength among the groups. SEM analysis revealed the formation of needle-like apatite crystals from the first day of SBF soaking, and EDS analysis indicated an increase in Ca, P, and Si elements. XRD and FTIR analyses confirmed the presence of apatite. This combination of additives improved the compressive strength and showed the good bioactivity performance of CO_3_Ap cement, making it a potential biomaterial for bone and dental engineering applications.

## 1. Introduction

Calcium phosphate cement is a bioactive bioceramic commonly studied as a bone substitute [1,2]. It has the ability to release calcium ions to its surface, providing a chemical bond to the hydroxyapatite structure of the bone, creating a good seal between the cement and bone [2,3,4]. The ability of this cement to form carbonate apatite (CO_3_Ap), with physical and chemical properties similar to bone mineral composition, its high solubility to be replaced with natural bone cells, and its capacity for osteogenesis, also make them a potential material for dentin-pulp regeneration. Recent research has shown that CO_3_Ap cement and Si-CaP in medical and dental applications have increased due to their unique properties, including biocompatibility and chemical similarity to natural bone [2,4,5]. In vivo studies showed that the material could create a physiological environment in the presence of blood from the pulp and induce dentinogenesis [3,6]. However, the most significant drawback of such bioceramics is the poor mechanical properties. This issue has been addressed through the development and incorporation of bioglass into the cement [7,8,9].

Silica in silica calcium phosphate composite (Si-CaP) can increase the material’s mechanical properties by establishing chemical bonding, which provides stability [6,9]. Si-CaP displayed a release of Si^4+^ and PO_4_^3−^ ions to its surroundings and contributed to osteoblasts’ maturation and hydroxyapatite formation on the surface layer [10,11]. On the other hand, calcium hydroxide [Ca(OH)_2_], a well-documented material in endodontics due to its superior antimicrobial property and the ability to induce dentinal bridge barrier on an exposed pulp, is also a good calcium ions provider. Ca(OH)_2_ works through the dissociation of Ca^2+^ and OH^−^ when in contact with water, elevating the alkalinity of the environment, resulting in a fatal effect on bacteria and stimulating the recruitment, proliferation, and activation of stem cells, undifferentiated cells, and odontoblast from the pulp to form reparative dentin. Long-term and meta-analytical studies have reported its superior use as a vital pulp therapy material [12]. Adding Ca(OH)_2_ to CO_3_Ap cement could benefit the cement’s biological properties. However, consideration must be taken as it could also weaken the mechanical properties of the cement.

The mechanical and bioactivity of CO_3_Ap cement are essential factors in the fabricating and synthesizing biomaterials. These materials have been found to promote bone growth and regeneration, making them ideal for use in various clinical applications [2]. Additionally, their ability to integrate with existing bone tissue has improved patient outcomes and reduced the need for additional surgeries. While CO_3_Ap and Si-CaP have been widely used in dental and medical applications due to their biocompatibility and similarity to natural bone, there have been concerns regarding their mechanical properties. Incorporating bioglass into the cement has been a solution to address this issue, but there is still a need to improve the mechanical properties and bioactivity of these materials for better clinical outcomes.

The bioactivity of these materials has been attributed to their compositional and surface characteristics. However, little is known about the interactions between the constituent minerals and the overall properties of the composites. Bioactive substances are brought into contact with the body’s fluids, and apatite begins to form on the surface of the bone. Evidence of a chemical bond between the material and the tissue can be seen in the formation of an apatite layer on the contacted tissue [13,14]. The release of Ca^2+^ and PO_4_^3−^ ions initiate apatite layer formation due to external pH changes [15]. In addition, the release of Ca^2+^ and PO_4_^3−^ ions plays an essential part in the tissue regeneration process and dentinogenesis or dentin-pulp regeneration [2]. The in vitro evaluation of cement’s ability to form an apatite layer can be carried out using a medium that simulates body fluid developed by Kokubo with ion concentrations similar to the body plasma [16,17].

We look at the potential properties of Ca(OH)_2_ to increase the Ca element of CO_3_Ap cement. However, concern also arises about compromising its mechanical properties. Therefore, the addition of Si-CaP could be a solution to overcome this issue while at the same time potentially enhancing the bioactivity. In this study, we investigate the mechanical property in terms of the compressive strength and bioactivity of CO_3_Ap cement incorporated with Si-CaP and Ca(OH)_2_ to form an apatite layer. We hypothesize that incorporating Si-CaP and Ca(OH)_2_ into CO_3_Ap cement will improve its compressive strength and show good bioactivity performance.

## 2. Materials and Methods

### 2.1. Preparation Specimen Groups

Vaterite and dicalcium phosphate anhydrous (DCPA) were utilized as a precursor for CO_3_Ap. Vaterite was synthesized according to the previous study (9). The DCPA (J.T. Baker Chemical Co., Phillipsburg, NJ, USA) and Ca(OH)_2_ (Emsure Millipore Corp., Burlington, MA, USA) powders were used without further processing. Si-CaP used in this study consists of 19.49% SiO_2_, 20.34% P_2_O_5_, 40.68% CaO, and 19.49% Na_2_O (in mol %) [18].

The cement powder was divided into one control group and five test groups. For the control group, a combination of only 60% DCPA and 40% vaterite were employed [19]. Each group has a similar composition for 60% DCPA and 30% vaterite but differs in the weight percentage of the addition of Si-CaP and Ca(OH)_2_, as shown in Table 1. All vaterite, DCPA, Si-CaP, and Ca(OH)_2_ powders were mixed homogeneously to obtain cement powder. The prepared powders specimen was manipulated with 0.2 mol/L of disodium hydrogen phosphate (Na_2_HPO_4_; pH 8.2) supplemented with _K_-Carrageenan (Sigma-Aldrich, Singapore, Singapore), a hydrosoluble polymer for the cement liquid to form a homogenous cement.

The compressive strength of the paste was evaluated by filling a Teflon mold with a diameter of 4 mm and a height of 6 mm following the ISO 7489:19868 (ISO 9917:1991 reference) standard. This standard involves preparing cement specimens in a mold and allowing them to be set under controlled conditions. The standard also specifies the compressive strength test’s testing conditions, including the specimens’ size and shape, the loading rate, and the temperature and humidity conditions during testing. These testing conditions are critical to ensure the accuracy and consistency of the test results, which is essential to ensure that the cement is reliable and suitable for its intended clinical applications. Powder and liquid phases had a liquid-to-powder (L/P) ratio of 0.4. The L/P ratio of 0.4 was the optimized L/P ratio determination through experiments and based on clinical expert assessment.

For the bioactivity evaluation, specimens were prepared in a Teflon mold with a diameter of 10 mm and a height of 2 mm, following the ISO 23317 standard [20]. This standard provides a method for testing the ability of materials to form a layer of apatite in simulated body fluid (SBF). The formation of apatite on the surface of an implant material indicates its ability to integrate with surrounding bone tissue and promote osseointegration. The test involves immersing the material in SBF for a specific time under controlled conditions. SBF is designed to mimic the ionic composition and pH of human blood plasma and promote apatite formation on the material’s surface. Following immersion in SBF, the material is analyzed for the presence of apatite using techniques, such as X-ray diffraction, scanning electron microscopy, or energy-dispersive X-ray spectroscopy. The analysis results are used to determine the extent and quality of apatite formation on the material’s surface.

Both ends of the mold were covered by glass slides and clamped by a metal clip. The molds were then stored in a container with distilled water to maintain 100% relative humidity and placed into an incubator at 37 °C for 72 h. After incubation, specimens were removed from the molds, immersed in 99% ethanol for 3 min, and dried in an oven at 80 °C for 3 h.

### 2.2. Compressive Strength Measurement

The set specimens were subjected to a compressive strength test to evaluate its mechanical strength. The compressive strength is evaluated by the maximum force applied when the specimen fractures and calculates the compressive strength value (C) in MegaPascals (MPa) using the below equation:$$C\;=4p/\pi d^{2}\;$$
where

*p* is the maximum force applied, in Newtons (N);

*d* is the measured diameter of the specimen, in millimeters (mm).

Ten specimens were used for each group, and the mean compressive strength value was used as data. The specimens were crushed using a universal testing machine (Lloyd Instruments Ltd., West Sussex, UK) with 5.6 N load stress at a 1 mm/min crosshead speed. The 5.6 N load stress was employed to apply force to the specimens and measure their resistance to compressive deformation. The inclusion criteria of the compressive strength specimen measurement were specimens that meet specific size and shape requirements, specimens free from visible cracks or defects, and specimens set after 72 h and ready for testing. Meanwhile, the exclusion criteria of the compressive strength specimen measurement were specimens with visible cracks or defects, specimens not fully set even after 72 h, and specimens not the correct size or shape for testing.

### 2.3. Preparation and Calculation of SBF

The SBF was prepared by Bogor Botanical Institute, Bogor, Indonesia, according to ISO 23317. The volume of SBF was calculated using the following equation: v_s_ = S_a_/10, where v_s_ is the volume of the SBF (mm^3^) and S_a_ is the apparent surface area of the specimen (mm^2^). The calculated SBF (21.89 mL) was kept in an enclosed bottle and maintained at 36.5 °C.

### 2.4. Bioactivity Evaluation Procedure

The group with the highest compressive strength was further evaluated for its bioactivity. The specimens were soaked into the SBF for 21 days, and the SBF solution was replaced twice weekly and evaluated at four different periods: 1, 7, 14, and 21 day(s) [11]. In each evaluation time, the specimens were taken from the SBF, gently rinsed with pure water, and then dried in a desiccator at room temperature for further analysis.

### 2.5. Scanning Electron Microscope and Energy Dispersive Spectroscopy (SEM-EDS)

The CO_3_Ap- Si-CaP -Ca(OH)_2_ specimen was coated with gold using a sputtering machine. Then, SEM at 15 kV accelerating voltage and EDS were used to examine the microstructure of the surface as well as the mass % of Ca, P, and Si elements (JEOL JSM-G510A, Tokyo, Japan).

### 2.6. X-ray Diffraction (XRD)

Set CO_3_Ap- Si-CaP -Ca(OH)_2_ cement specimens were ground to obtain fine powder for XRD analysis. Specimen powder was examined by XRD machine (D2 Phaser, Bruker Corp., Billerica, MA, USA) in the range of 3°–50° in 2 theta  (θ) using CuK*a* (λ = 0.15405 nm) radiation operated at 40 kV of tube voltage and 40 mA of tube current.

### 2.7. Fourier-Transform Infrared (FTIR)

The evaluation of the FTIR (Spectrum 100, Perkin Elmer Inc., Shelton, CT, USA) was performed using the KBr method. The KBr and CO_3_Ap- Si-CaP -Ca(OH)_2_ fine powder at a 200:1 ratio was mixed homogenously, and the compacted pellets were put in a stainless steel die. A spectral resolution of 4 cm^−1^ and a range of wavelengths from 400–4000 cm^−1^ were used to investigate the chemical structural changes.

### 2.8. Statistical Analysis

KaleidaGraph 4.1 (Synergy Software, Reading, PA, USA) was used to perform one-way factorial ANOVA with Fisher’s LSD method as a post hoc test on data relevant to compressive strength. The level of statistical significance was chosen at *p* < 0.05.

## 3. Results

### 3.1. Compressive Strength Evaluation

Figure 1 summarizes the compressive strength of the specimens after setting in 100% relative humidity at 37 °C for 72 h. The compressive strength of the cement specimens varied significantly among the six groups. Group 2 exhibited the highest strength at 23.97 ± 4.46 MPa; this difference was statistically significant compared to all other groups (*; *p* < 0.05). This indicates that the properties of the cement in Group 2 were notably different and most resistant to compression. On the other hand, Group 3 had the lowest compressive strength at 11.05 ± 5.92 MPa and this difference was also statistically significant compared to the other groups, indicating that the cement in this group was the least resistant to compression. Additionally, it is noteworthy that there was a statistically significant (**; *p* < 0.05) difference between Group 1 and Groups 3 and 4, with Group 1 exhibiting higher compressive strength than Groups 3 and 4, indicating different properties of the cement in Group 1 compared to those in Groups 3 and 4.

Regarding the comparison between the control group and Group 1, the results showed a slight improvement in compressive strength in Group 1 compared to the control group. However, the difference between the mean values of the two groups was not statistically significant (*p* > 0.05). This suggests that the addition of Ca(OH)_2_ alone did not significantly improve the compressive strength of the cement. Nonetheless, the slight improvement observed in group 1 could indicate the potential of CO_3_Ap to enhance the compressive strength of the cement when combined with other materials, such as Si-CaP and Ca(OH)_2_, as observed in Group 2. Overall, the results demonstrate that incorporating Si-CaP and Ca(OH)_2_ did not universally improve the compressive strength of CO_3_Ap cement. Only Group 2 showed a significant enhancement, while Groups 3–5 resulted in a loss of compressive strength.

### 3.2. SEM-EDS Analysis

In order to assess the bioactivity of the CO_3_Ap cement, the formation of the apatite layer was observed using SEM. The SEM images showed that the apatite layer formation began with needle-like crystal growth in both the control and bioactivity specimens. Figure 2A,B show the surface of the control specimen before soaking in SBF, while Figure 2C,D depict the apatite crystal growth after one day of exposure to SBF. As expected, no crystal growth was observed on the control specimen before being exposed to SBF, as apatite formation requires the cement to be in contact with SBF. The SEM images demonstrate that the bioactivity specimen had a denser arrangement of apatite crystals than the control specimen after one day of exposure to SBF.

After seven days of soaking in SBF, the apatite crystal growth was faster in both specimens, as shown in Figure 2E,F. However, the arrangement of the apatite crystals was less regular than in Figure 2C,D. The irregular crystal structure of the apatite formed due to multiple crystal growth mechanisms and the transformation of calcium phosphate [21] can explain this phenomenon. Figure 2G,H show that the apatite crystal fragments were still merging and not yet conjugated.

Finally, after 21 days of SBF exposure, the cross-section SEM revealed that the apatite layer grew denser and more homogeneous (Figure 2I,J). However, the formation of the apatite layer was not uniform across the surface, as evidenced by the black regions in the SEM images. There was no obvious interface, and the growth of the apatite layer depended on the extended soaking period.

Figure 3 summarizes the elemental component of set CO_3_Ap- Si-CaP -Ca(OH)_2_ cement after being soaked into SBF in four different periods of exposure (1, 7, 14, 21 days) that were analyzed using EDS. The measured area was 30 microns in diameter and located at the specimen’s center. The mass% of Ca element increased over time. The average mass% of Ca element was 17.34%. The most significant increase in Ca element mass% was 18.55%, which occurred both before and after the specimens were soaked in SBF. In this study, the average P element mass% was 6.67%, and the most significant increase in P element mass% was 8.83%, which occurred between 14–21 days of soaking. However, the P element mass% did not exhibit a constant upward trend as the graph of Ca element. The mass% of P element decreased in one day and 14 days, respectively. In the case of Si, the average Si element was 4.45%, and the highest Si element concentration was 7.82% after 21 days of soaking.

### 3.3. FTIR Analysis

The FTIR spectra of the set cement are summarized in Figure 4. FTIR analysis revealed that all specimen spectra differ before and after soaking in SBF. The CO_3_^2−^ bands were detected at 863–879 cm^−1^ and 1427–1445 cm^−1^ in all specimens. In addition, the P-O symmetric stretch was also seen at 1027–1040 cm^−1^. These bands indicated CO_3_^2−^ and PO_4_^3−^ ionic substitution with the apatite structure. Due to the incorporation of Si-CaP, the spectra of set cement additionally showed the Si-O-Si vibration bands at 465 cm^−1^ and 1107 cm^−1^.

### 3.4. XRD Analysis

The XRD spectra analysis (Figure 5) revealed that the control and bioactivity specimens underwent a phase transformation over time. The vaterite peaks in the range of 2θ = 27.1–27.3, 32.9–33.1, and 43.5–43.7 progressively decreased with increasing SBF soaking time, indicating the dissolution of the vaterite phase. Meanwhile, the peaks corresponding to an HA-like phase appeared and increased in intensity over time in the range of 2θ = 26–26.1 and 32–32.04. After seven days of soaking, the vaterite peaks disappeared completely from the XRD spectra, and the presence of the HA-like phase was dominant. These results demonstrate that the CO_3_Ap cement is capable of undergoing a phase transformation to form an apatite phase in both the control and bioactivity specimens over time.

## 4. Discussion

Our previous study demonstrated that 40% vaterite and 60% DCPA mixed with various sodium phosphate solutions fully transformed to CO_3_Ap with good osteoconductivity. However, the mechanical strength of the cement still needs to be improved to gain good clinical application [19]. In this investigation, 7% Ca(OH)_2_ and 3% Si-CaP improved the compressive strength and showed good bioactivity. However, it is important to note that the cement with the additives above has good bioactivity but is not necessarily enhanced. It is well established that the Si-CaP possesses advantageous physiochemical and bioactivity features [22]. When DCPA, vaterite, and Ca(OH)_2_ were mixed with the 0.2 mol/L of Na_2_HPO_4_, the powder dissolved and supplied Ca^2+^, PO_4_^3−^, and CO_3_^2−^ ions. These ions became supersaturated and precipitated into CO_3_Ap.

On the other hand, Si-CaP contributes to forming a network; the Si-OH from Si-CaP binds calcium supplied by DCPA, vaterite, and Ca(OH)_2,_ forming a more dense structure, thereby increasing the compressive strength of this set cement. Additionally, incorporating Si-CaP into the cement has been shown to increase its compressive strength, making it a better option for clinical use. This is because the advantageous properties of Si-CaP not only enhance the mechanical properties of the cement, but also improve its bioactivity, making it more biocompatible and suitable for use in the body.

The ability of a material to cause a particular biological response is referred to as its bioactivity [23,24]. This response includes the production of hydroxyapatite layers on tissue surfaces. In vitro bioactivity evaluation helps determine whether a new bioactive substance can form bonds between tissue and material and serves as preliminary research for tests performed in vivo [20].

Our study showed that despite no exposure to SBF, the needle-like apatite crystal growth surrounding the globular structure can still be observed. A dissolution-precipitation reaction may be seen here, as evidenced by the interlocking of the precipitated apatite crystal structure [21,25,26]. The CO_3_^2−^, Ca^2+^, and PO_3_^4−^ ions dissolved and precipitated to form CO_3_Ap crystals. Later on, the CO_3_Ap precipitates and interlock with one another to form a more compact CO_3_Ap arrangement [2].

As a result of contact between the CO_3_Ap- Si-CaP -Ca(OH)_2_ cement with body fluid (in this study SBF), a dissolution-precipitation process took place as observed in the formation of the apatite layer (Figure 2C,D). The following describes each of these chemical steps: (1) surface de-alkalization exchange with H^+^, which leads to an increase in the local pH; (2) formation of Si(OH)_4_; (3) re-polymerization of Si(OH)_4_ to form SiO_2_ rich layer; (4) precipitation of Ca and P ions; and (5) crystallization of CaO-P_2_O_5_ layer to form crystalline apatite structures [23,27].

The transformation of calcium phosphate into apatite resulted from numerous crystal growth mechanisms and the reconstruction of apatite crystals [28]. This corresponded to the irregular arrangement of apatite crystal, which emerged after seven days of soaking in SBF. With a longer SBF exposure time, the crystal arrangement of the apatite becomes denser. However, this arrangement does not appear to have formed uniformly across the surface of the specimen, even after it was subjected to SBF for 21 days. Apatite layer development shows evidence of an apparent dark non-collagenous interfacial zone (Figure 2G,H). The presence of an interfacial zone is consistent with previous studies showing the formation of the apatite layer [29,30]. Apatite crystals were formed by apatite precipitation in an aqueous solution at physiological pH and room temperature [31].

FTIR and XRD analyses provide evidence that apatite crystals are present. FTIR evaluation verifies the presence of C-O bands and a P-O symmetric stretch, which point to the ionic substitution of CO_3_^2−^ and PO_4_^3−^ within the apatite structure. In line with the FTIR results, the XRD result confirmed that apatite increases with the overexposure time of SBF. The rising Ca and P element is another essential indicator in bioactivity evaluation. Before and after soaking the specimen in SBF, there was an increase in the Ca element mass% that ranged from 13.49% to 18.55%. Besides Ca ions, P and Si ions contribute to mineralised tissue regeneration and induce odontogenic differentiation to form dentinal bridge formation [8].

This study employs Si-CaP and Ca(OH)_2_ to improve the compressive strength and shows good bioactivity of CO_3_Ap cement but is still limited in scope and design. A limited number of variables were evaluated, and the timeframe over which the compressive strength and bioactivity of the CO_3_Ap cement supplemented by Si-CaP and Ca(OH)_2_ were assessed was short. In addition, the studies were performed on a microstructural level only, leaving out the macro- and micro-scale biological factors that govern the properties of the cement in the physiological environment. Therefore, there is a need to further study the effects of these additives of Si-CaP and Ca(OH)_2_ on the in vitro and in vivo study of CO_3_Ap cement.

## 5. Conclusions

The current study has demonstrated that incorporating Si-CaP and Ca(OH)_2_ into CO_3_Ap cement significantly enhances its compressive strength. Furthermore, the resulting CO_3_Ap- Si-CaP -Ca(OH)_2_ cement exhibits good bioactivity, as evidenced by apatite formation on its surface, as well as an observed increase in the average mass% of Ca, P, and Si elements, and the early formation of apatite crystals following immersion in the SBF solution. Further studies incorporating a control group are necessary to confirm its enhanced bioactivity. Overall, our findings suggest that CO_3_Ap- Si-CaP -Ca(OH)_2_ cement is a promising biomaterial for use in bone and dental tissue engineering applications.

## Figures and Tables

**Figure 1 materials-16-02071-f001:**
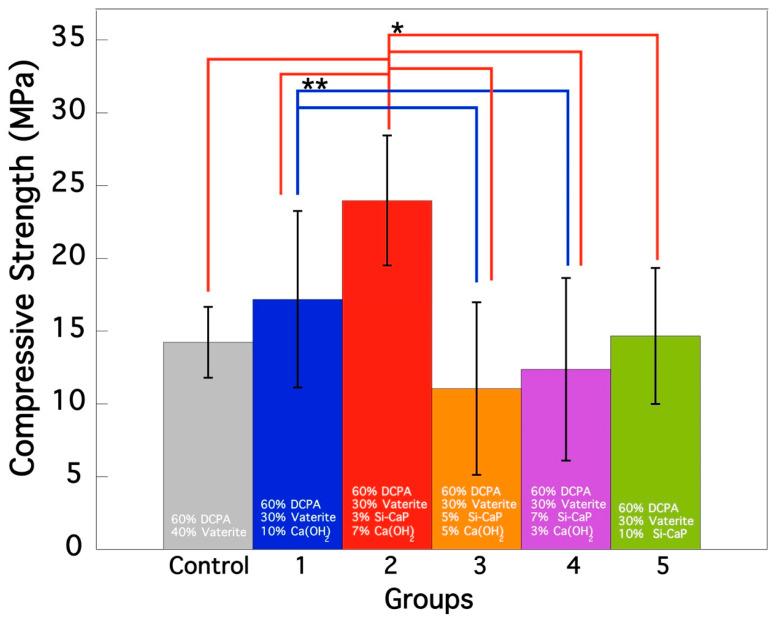
Compressive strength evaluation of CO_3_Ap- Si-CaP -Ca(OH)_2_ cement specimens after 72 h of treatment. Ten specimens of each group were measured for compressive strength, and error bars indicate the standard deviation. Symbols of * and ** were showed statistically significant between the groups.

**Figure 2 materials-16-02071-f002:**
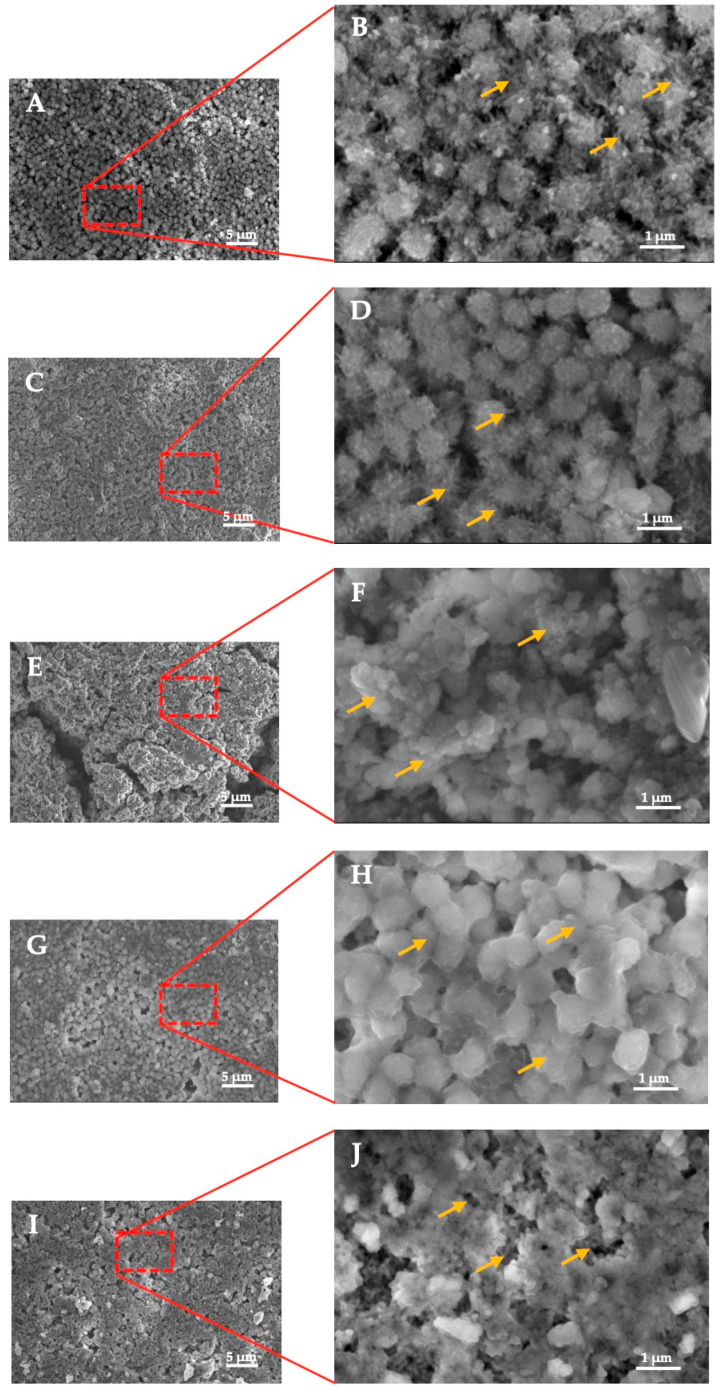
(**A**,**B**) control specimen; (**C**,**D**) 1-day exposure to SBF; (**E**,**F**) 7 days exposure to SBF; (**G**,**H**) 14 days exposure to SBF; (**I**,**J**) 21 days exposure to SBF. Noted: The red rectangle is the location of image magnification; the yellow arrow indicates the apatite crystal’s growth and the apparent interface of the apatite layer.

**Figure 3 materials-16-02071-f003:**
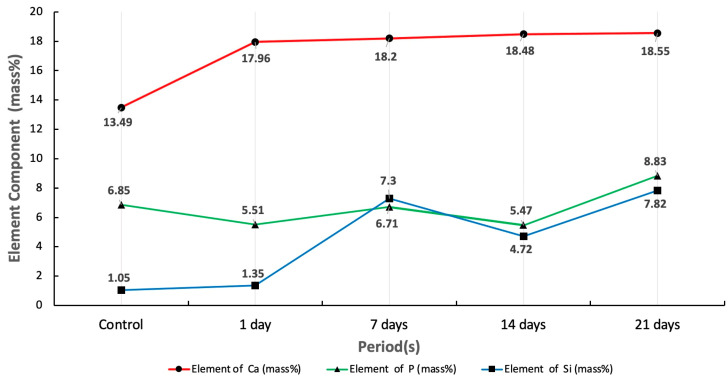
Evaluation of Ca, PO and Si elements before and after soaking into SBF for 1, 7, 14, and 21 day(s).

**Figure 4 materials-16-02071-f004:**
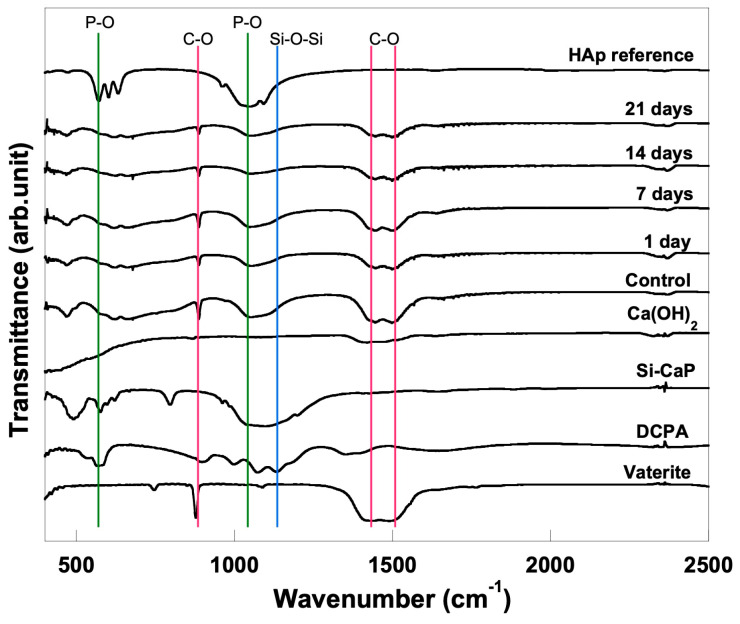
FTIR spectra of CO_3_Ap− Si−CaP −Ca(OH)_2_ set cement before and after soaking into SBF for 1, 7, 14, and 21 day(s).

**Figure 5 materials-16-02071-f005:**
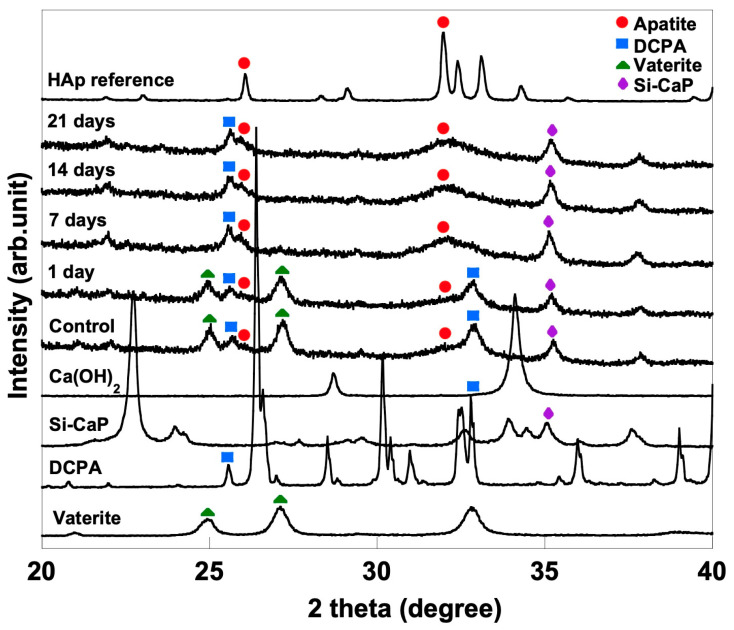
XRD patterns of CO_3_Ap- Si-CaP -Ca(OH)_2_ set cement before and after soaking into SBF for 1, 7, 14, and 21 day(s).

**Table 1 materials-16-02071-t001:** The powder composition specimens.

Groups	Powder Compositions	Additional Powder
Control	60% DCPA and 40% vaterite	
1	60% DCPA and 30% vaterite	0% Si-CaP and 10% Ca(OH)_2_
2		3% Si-CaP and 7% Ca(OH)_2_
3		5% Si-CaP and 5% Ca(OH)_2_
4		7% Si-CaP and 3% Ca(OH)_2_
5		10% Si-CaP and 0% Ca(OH)_2_

## Data Availability

Not applicable.

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
