# Peer review of "Bioactive Carbonate Apatite Cement with Enhanced Compressive Strength via Incorporation of Silica Calcium Phosphate Composites and Calcium Hydroxide"

_materials, 2023, doi:10.3390/ma16052071_

Round 1

Reviewer 1 Report

The manuscript investigates the mechanical and bioactivity of carbonate apatite cement with silica calcium phosphate composites and calcium hydroxide additions. The manuscript should be improved further concerning the following comments before publication.

1.        Abstract should be revised. It is a little bit confusing. The carbonate apatite is synthesized by dicalcium phosphate anhydrous and vaterite powder, but it is not mentioned. Group 2 (34th line) may not be necessary.

2.        Materials and methods can be improved. Line 114, control group is 60% DCPA and 40% vaterite. In Fig. 3 and thereafter, the control is the best composition. This should be consistent. Line 117, what is the hydrosoluble polymer? Line 118, the liquid-to-powder ratio of 0.4 is used, why?

3.        Figure 1 may be deleted. Is it necessary to keep Fig. 1?

4.        Line 143, why 5.6 N load stress is used?

5.        In Figure 2, how do you perform the statistical analysis? Some more information should be given in the “Materials and Methods” section. What is the symbol “*” … implied?

6.        Figure 3, the needle-like crystal is not clear at this magnification? Replace with a higher resolution.

7.        Figure 4, the results are mainly from EDS. What is the measured area?

8.        Line 372, I may not agree with “7% Ca(OH)2 and 3% Si-CaP” improved the mechanical strength and bioactivity, because only one set of sample was evaluated. At least, you have to compare with the original one (60% DCPA and 40% vaterite ) to make this statement.

Author Response

We have responded to all the reviewer comments; please kindly check the attached file.

Reviewer 2 Report

Dear Authors

I am pleased to review your manuscript. Over all of this manuscript is appreciated. I have only few suggestion to improve. 

About the 2.1 preparation specimen group if possible, can you add the table of the sample composition the reader will more understand.

Please add more reference about ISO standard in this menuscript.

The result of 3.2 SEM-EDS analysis, how can you collect the data of Figure 4 if it from EDS the data should be collect from many point, the graph in figure 4 please include error bar.

Author Response

(The authors gave the same response as above.)

Reviewer 3 Report

Dear Authors

I have read your manuscript with great interest. The idea of incorporating Si-CaP and Ca(OH)2 sounds good; however, considering the way you carried out the experiments, you cannot asseverate, as you do in the conclusions and even in the title, that there is an enhancement of mechanical properties and bioactivity with this incorporations.

You investigated two aspects of the materials. The first was the compressive strength, and to be able to say that there was an enhancement in this property, it is necessary to know the values of compressive strength in the base material without adding the other products (Si-CaP and Ca(OH)2). Even though in lines 113-114 you mention that this control group was prepared, there are no results for this group, neither in the results text (lines 176-182) nor in figure 2 and much less address this in the discussion.

The same problem also occurs when investigating bioactivity. In lines 152-153, you mention that the group with the highest mechanical strength was further evaluated for its bioactivity, but the control group was not used to compare the results of the study at different times of exposure to SBF, with none of the techniques used (SEM EDS, XRD, FTIR).

So how do you know if there was an enhancement of the compressive strength or in the bioactivity? What if these products instead decreased both? It is essential to have information on the material without additives exposed to the same conditions as the experimental groups.

 Also, regarding what you call "mechanical properties" or "mechanical characteristics," as you mention several times in the document, the reality is that you only investigated concretely and specifically the "compressive strength" this is only one of the various mechanical properties that a material exhibit, the manuscript should be limited to this term only.

 Regarding this test, there is information that must be clarified in order to understand precisely how the compressive strength test was performed. For example: Why do you say "at least ten specimens"? Please mention exactly how many were in each group and why?, and what were the inclusion and exclusion criteria of the specimens? Why 10? And explain how you could record the newtons needed to destroy eachone of the disks and obtain averages if, in all cases, you applied a constant force of 5.6N.? It is also necessary that you indicate the formula that you used to obtain the MPa.

On the other hand, it is also necessary to do a spelling and grammatical review and unify some terms in addition to the term "compressive strength" (previously discussed). For example:

 Line 33, 145: "simulation" should be "simulated"

Line 39: what do you mean by "biocompatible susceptibility"?

Line 74: what do you mean by "construction biomaterials"?

What is the difference between "CO3Ap" (line 24) and "CO3Ap cement" (line 27)? In line 49, you specify that "CO3Ap" is CARBONATE APATITE. But in line 46 you write, "Calcium phosphate cement." In all cases, do you mean the same thing? It's confusing.

Figure 1 is unnecessary. Do you really think that figure 1b is helpful?

These are just a few examples, but there are more that need to be revised

Author Response

(The authors gave the same response as above.)

Round 2

Reviewer 1 Report

The authors have replied the comments. No further suggestions. 

Author Response

Please kindly check the file attached. Thank you.

Reviewer 3 Report

Dear Authors

Thank you very much for the effort to correct your manuscript. I must say that it has gotten better; however, I insist again that the biggest problem is that considering the way you carried out the experiments, without a control in the bioactivity evaluation, you cannot assert, as you do in the conclusions and even in the title, that there is an enhancement of the bioactivity with this incorporations.

I am happy to read that you agree on the importance of the control group, and I am even more pleased to see that you added it to the compressive strength tests. With that reference point, you could realize that only two of the proposals cause an enhancement while the other 3 (groups 3,4 and 5) cause a loss in compressive strength. These results no longer make your title or conclusion correct in the part where you conclude that: …" By incorporating Si-CaP and Ca(OH)2, the compressive strength of CO3Ap cement were improved…." Since this was not true in general, apparently only two options did and perhaps not even two, since the difference between the control group and group 1 does not seem to be significant (it is not possible to know because you do not indicate anywhere if the difference between control and group 1 was significant), this only leaves group two with an "enhancement" while of the other four, three of them worsened it significantly (groups 3, 4 and 5) and in one of them, it was not a significant "enhancement" (group 1).

All this gives more evidence that you cannot prove an "enhancement of bioactivity" either, since in this case (bioactivity), there was not even a proper control group where the material without additives was subjected to the same conditions to distinguish whether or not there is an "enhancement" therefore, and although you mention that you will take it into account for future experiments, this does not make it possible for you to ensure that there was an enhancement in bioactivity as you say in the title and later in the discussion: (lines 408,409)…improved the compressive strength (I agree, is correct) and bioactivity (this is not possible to now and therefore cannot be assured) as I told you previously how do you know if there was an enhancement or a diminish in of the bioactivity? What if these products instead decreased it? It is essential to have information on the material without additives exposed to the same conditions as the experimental groups; since you do not have them, you only have two options left, repeat the experiments or make the necessary adjustments in the manuscript so as not to say that there is an "enhancement" maybe you can report that this material has good bioactivity but not that it is "enhanced."

For all this, it is incorrect that in section 3.2, line 237, you compare with the "supposed" control since that is not an appropriate control. As you say, ....it is the same specimen prior to soaking in SBF....., it is evident that at that moment, the formation of crystals will not be observed because, as you also say in lines 87-89,....for apatite to form, the cement must be in contact with SBF.....

On the other hand, you still have to do a detailed analysis of the texts; there are still repeated ideas in some sections, for example (but there are more):

1.- lines 51-52: ….The previous study has shown promising results of CO3Ap used in dental fields…

It's almost the same idea as

lines 73-75: …..CO3Ap cement and Si-CaP have been 73 widely used in various dental and medical applications because of their high biocompatibility and chemical similarity to natural bone…

2.- Lines 54-55:  …. However, the most significant drawback of this bioceramics is the poor mechanical properties; this issue has been addressed through the development and incorporation of bioglass into the cement…

It's almost the same idea as

Lines 75-76: ….However, the mechanical properties of these materials are typically low, and the bioactivity is relatively low as well. Therefore, these composites' mechanical properties and bioactivity need to be improved….

3.- Lines 409-410: …. the Si-CaP possesses 409 advantageous physiochemical and bioactivity features ….

It's the same as

Lines 417:…. Si-CaP possesses advantageous physiochemical and bioactivity features….

Regarding my previous comment about explicitly using the term compressive strength instead of mechanical, it is true that you corrected some. However, the title remains the same, even in subsections "2.2. Mechanical strength measurement" and the "3.1 Mechanical strength evaluation". In the text, there are others where you continue to use the term mechanical impropriety.

Regarding my comment about unifying the term "SBF" it is true that you have already corrected most of them. However, others continue to appear in the text that should only be abbreviated, for example, in lines 167 and 169.

The results section in the paragraph 202-216 must be restructured; the way it is written is very confusing. For example, line 204: …Group 2 had the highest compressive strength… then in line 208: ….This indicates that the cement in group 2 was the most resistant to compression… and then in line 209: … was able to withstand higher compressive forces than the other groups….

You're basically saying the same thing 3 times.

On the other hand, it is also necessary to do a spelling and grammatical review as there are texts that are not clear; some examples are (but there are more):

Line 51: …The previous study…    (singular, but there are three references)

Line 53: … The material could set physiological conditions...

Line 205, 207, 213, about …statistically significant… the term is not being used well; you should mention that the difference between groups was or was not statistically significant.

Author Response

(The authors gave the same response as above.)
